# Wearable Collar Technologies for Dairy Cows: A Systematized Review of the Current Applications and Future Innovations in Precision Livestock Farming

**DOI:** 10.3390/ani15030458

**Published:** 2025-02-06

**Authors:** Martina Lamanna, Marco Bovo, Damiano Cavallini

**Affiliations:** 1Department of Veterinary Medical Sciences (DIMEVET), University of Bologna, 40064 Bologna, Italy; martina.lamanna5@unibo.it (M.L.); damiano.cavallini@unibo.it (D.C.); 2Department of Agricultural and Food Sciences (DISTAL), University of Bologna, 40127 Bologna, Italy

**Keywords:** precision livestock monitoring, sensor technology, animal behaviour tracking, smart farming, digital dairy management

## Abstract

Wearable collar technologies are revolutionizing dairy farming by providing real-time insights into cow health, behaviour, and productivity. These collars, equipped with sensors, allow farmers to monitor key parameters non-invasively, improving animal welfare and farm efficiency. Despite their potential, challenges such as limited battery life and high costs hinder their broader adoption. Future advancements should focus on integrating multiple sensors into energy-efficient designs, reducing costs, and simplifying management. This study emphasizes the pivotal role of smart collars in precision livestock farming, enabling sustainable practices and optimizing herd management. By addressing existing limitations, these technologies can drive a significant transformation in livestock farming, aligning with global goals for sustainability and innovation.

## 1. The Dairy Sector in the Global Economy

The dairy sector plays a central role in the global agricultural economy, contributing not only to food production but also to economic development in many parts of the world [1]. In the current livestock context, the challenges related to environmental sustainability, production efficiency, and animal welfare are becoming increasingly pressing. Worldwide bovine and buffalo milk production represented 96% of total milk production in 2021, totalling 884 million tonnes. This marked a 58% increase (i.e., 328 million tonnes) compared to 2000 [1]. The growing demand for high-quality dairy products, along with the need to reduce environmental impact [2,3,4,5] and comply with increasingly stringent animal welfare regulations, makes a radical transformation of production systems indispensable [6]. This push for change finds strong support in the European Union, which, through targeted policies and funding programs such as the Green Deal, the “Farm to Fork” strategy, and the Common Agricultural Policy (CAP), recently renewed for the 2023–2027 period, promotes the adoption of innovative and digital technologies in the agricultural sector.

## 2. The Role of Precision Livestock Farming and Its Global Spread

In the context of addressing the challenges faced by the dairy sector, including environmental sustainability, production efficiency, and compliance with stringent animal welfare regulations, precision livestock farming (PLF) represents a multidisciplinary and interdisciplinary concept based on the use of digital technologies for continuous and real-time monitoring of a wide range of data related to animals and the surrounding environment [7,8]. PLF aims to enhance environmental, economic, and social sustainability by optimizing the health and welfare of individual animals, thereby improving the overall efficiency of livestock management [7,8,9]. To achieve these goals, it relies on sensors that collect information on parameters such as physical activity, geographic location, body temperature, feeding, and drinking behaviour, as well as many other vital indicators [10]. The data collected by the sensors are then processed and interpreted through predictive algorithms, which analyze the information and identify anomalies or changes that could indicate health or welfare issues [8,11,12]. Digital interfaces allow farmers to visualize the collected data and integrate them with other animal-related information, generating specific insights regarding health, behaviour, production, and the environment. This supports the decision-making process, which may be performed by the farmer or fully automated, based on scientific evidence and predictive data analysis [8,11,12]. This process enables proactive and individualized management, reducing the need for manual interventions—such as physically inspecting each cow for signs of illness, manually identifying cows in heat for breeding purposes [8], or adjusting feed rations based on observed behaviour [12]. Instead, automated systems can detect early signs of disease through changes in rumination patterns, trigger alerts for optimal insemination timing based on activity data, and provide recommendations for adjusting feed composition based on real-time consumption metrics, significantly improving the overall efficiency of farm operations [8,11,12].

The time span from 2005 to 2022 highlights the significant worldwide scientific interest in the adoption of PLF sensors in cattle farming [13].

As PLF technologies continue to evolve, wider adoption of these solutions is expected, not only in countries that are already advanced in this sector but also in new regions, further contributing to more efficient, sustainable, and responsible livestock management on a global scale. In fact, an aspect that today appears unequivocally is that the adoption of PLF systems leads to an increase in the sustainability of production [14,15,16], hence the interest in continuing to develop new technologies applicable in the livestock sector. In such a dynamic and constantly developing context, the technological system based on PLF techniques that has been most investigated and used is that of devices that can be worn by animals. In the dairy cow sector, this system, in addition to being the most investigated, is probably the most widespread and used by farmers to date as a daily support tool in herd management.

## 3. PLF Collars: Typologies and Functions

The sensors used for monitoring and recording data on animals have become indispensable (e.g., accelerometers), as they overcome the logistical challenges of direct animal observation. These sensors are employed both in commercially available devices and in research prototypes. Data loggers, which are often integrated into these devices, can store data on-board for later retrieval or transmit such data in real time via wireless technologies. Multi-sensor data recorded simultaneously at high frequency—such as the combination of accelerometer and gyroscope data—provide an opportunity to obtain detailed information related to behavioural parameters by combining and analyzing multiple sensor outputs [17]. The sensors used in PLF can be classified into different categories depending on their position relative to the animal and the parameters they monitor. Each type of sensor plays a fundamental role in collecting data useful for management optimization. They can be classified as “on-animal” sensors, “near-animal” sensors, and “from-animal” sensors [18]. On-animal sensors are integrated into data loggers directly attached to the cow (such as collars, ear tags, and reticula-ruminal boluses). Near-animal sensors are positioned in the immediate vicinity, within farm structures (e.g., sensors detecting temperature, humidity, air quality, and concentrations of harmful gases, as well as microphones and cameras). From-animal sensors collect data indirectly (e.g., flow meters can be installed in milking robots to analyze the amount of milk produced) [18]. Among the most commonly used sensors in PLF systems, particularly in dairy farm management, are activity sensors (e.g., accelerometers to monitor movement and lying behaviour), sensors with a Global Positioning System (GPS) antenna, sensors with a Radio frequency identification (RFID) tag, pressure sensors, intra-ruminal sensors, intra-vaginal devices, environmental sensors (e.g., temperature and humidity sensors to monitor barn conditions), microphones, magnetometers, gyroscopes, cameras and thermal cameras, thermometers, and heart rate monitors [11,19]. On-animal sensors can be positioned on different parts of the animal’s body, depending on the position that is optimal for the monitoring of specific behaviours. Common areas include the neck (i.e., collars), jaw, ear (using ear tags), forehead (e.g., pressure sensors or microphones), legs (pedometers), upper back, nose, and rumen (i.e., ruminal boluses).

Accelerometers mounted on collars in the neck are widely used to detect cow activity, body posture, and head movements [20,21,22,23,24]. Sensors for localization (based, for instance, on GPS antenna) are often combined with accelerometers to track animals’ positions and monitor the spatial dispersion of the herd [25,26,27]. Gyroscopes and magnetometers, also mounted on the neck, provide additional information about angular movements and head orientation [28,29,30,31,32,33,34,35]. Additionally, vocalizations can be recorded by a microphone attached to the collar [36]. Cow identification can be effectively carried out using RFID tags integrated into the collar [37].

Among these, collar-mounted sensors have emerged as one of the most versatile and widely adopted tools for dairy cow monitoring, offering a unique combination of advantages for tracking behaviour, activity, and physiological parameters. The objective of this study is to provide a comprehensive overview of the current state of knowledge regarding collar-mounted sensors for dairy cows, summarizing their functionalities and applications in behavioural and physiological monitoring. By consolidating existing research, this study aims to clarify the specific contributions of collar-mounted sensors to precision dairy management and identify potential gaps in their current use. Collars were chosen because they offer a comfortable fit, ensuring minimal stress and disruption to the animal’s natural behaviour. Furthermore, their positioning on the neck provides a stable platform for accurate sensor readings across various activities. While previous reviews [10,38,39,40,41] have provided valuable insights into commercially available and validated sensor technologies for dairy cattle welfare assessment, this study adopts a more focused approach. Specifically, our review refers exclusively to collar-mounted devices, offering a detailed and precise overview of both currently implemented and potential sensor technologies for behaviour, health, and productivity monitoring in dairy cows.

This focused perspective aims to complement existing broader reviews by providing readers with targeted insights into a specific and widely adopted sensor platform, contributing to a more detailed understanding of collar-mounted technologies in the context of precision livestock farming.

## 4. Materials and Methods

The work presented in this paper is part of a broader study aimed at laying the groundwork for the development of advanced collar-based systems in PLF, focusing on improving sensor integration to enhance data collection, accuracy, and multi-sensor functionality. To ensure a comprehensive understanding of the state of the art, a systematized literature review was adopted.

The review process began with a detailed search conducted across three major academic databases: Scopus, Web of Science, and Google Scholar. These platforms were selected for their extensive coverage of scientific and technical literature. The search strategy employed a combination of carefully selected key terms and synonyms: “precision livestock farming”, “PLF”, “collar”, “dairy cow”, “dairy cattle”, “sensors”, and “review”. Boolean operators (AND, OR) and database-specific filters (e.g., publication type, language, and peer-reviewed sources) were applied to refine the results and ensure high-quality, peer-reviewed articles were included. The exact search formula, for example, in Scopus, was as follows:

(“precision livestock farming” OR “PLF”) AND (“collar”) AND (“dairy cow” OR “dairy cattle”) AND (“sensor”) AND (“review”).

While searches of the Web of Science database were performed broadly across available indexes, we did not restrict our search specifically to the SCIE or SSCI sub-databases. This choice was made to ensure broader coverage and inclusivity of relevant publications across disciplines. In addition to database searches, a snowball sampling method was employed to identify additional relevant documents. This method involved examining both the reference lists of key studies and citations of those studies in subsequent research. The snowballing process was conducted iteratively, starting with foundational papers and expanding outward to capture significant advancements and seminal works in the field. Specific criteria guided this phase to avoid overly broad or tangential results, ensuring relevance to the study objectives.

The inclusion criteria focused on studies addressing the application of collar-mounted sensor technology in dairy cattle, particularly those related to behaviour monitoring, health management, and productivity assessment. Articles were excluded if they did not specifically address collar-mounted systems, if they focused solely on other sensor types, or if they fell outside the scope of PLF applications. In total, 55 articles were selected for inclusion in the review, covering a time span from 1997 to 2025, and representing a diverse spectrum of research on collar-mounted sensors. The extracted literature was analyzed to identify key trends, challenges, and opportunities in the field. Special emphasis was placed on understanding the functionality and practicality of collar-mounted sensors, given their ease of attachment, non-invasive nature, and consistent monitoring capabilities with regard to essential dairy cow behaviours. The findings were then synthesized to provide a holistic view of the current state of technology and its potential applications for improving livestock management practices.

## 5. Sensors Integrated in Wearable Collars

In the following subsections, the different types of sensors that can be mounted on a collar for cattle will be individually reviewed and examined to highlight the specific functionalities and applications in PLF. Our review focuses exclusively on collar-mounted sensors, offering a detailed analysis of current and potential technologies for monitoring in dairy cows.

This work is part of a broader study aimed at establishing the foundation for developing advanced collar-based systems, with an emphasis on improving sensor integration for enhanced data accuracy, collection, and multi-sensor functionality. The order of presentation is arbitrary and does not reflect any hierarchy of importance.

### 5.1. Accelerometer

One of the most useful types of information in PLF approaches pertains to the movement of individual animals. To achieve this, a technique typically employed relies on devices capable of determining the acceleration to which a mass is subjected. Accelerometers are used to remotely monitor animal behaviour by recording acceleration along three axes (usually X, Y, and Z). Most accelerometers use capacitive or piezoelectric position sensors. Capacitive sensors operate based on changes in electrical capacitance that occur when the distance between the two plates of a capacitor is altered; one plate is attached to the mobile mass, while the other is fixed to the device structure. Piezoelectric sensors, on the other hand, work by detecting the migration of charges that occurs when certain materials are subjected to external stress. Using a specific circuit, this variation is converted into a potential difference proportional to the displacement [36]. These devices are usually powered by an integrated battery that provides the necessary voltage to the various components. Many sensors equipped with 3D accelerometers support microcontrollers capable of performing data pre-processing, such as averaging the acceleration values collected over a predefined period and storing or transmitting only the average data. Information can be temporarily stored on the device itself, but a data transfer system, usually via radio, is generally needed. Data transfer through a cable connected to a computer are typically limited to experiments involving a few animals and conducted over short periods [36].

Once connected to the animal, the accelerometer tracks its movements, generating data on accelerations along the monitored axes [42]. However, before the device can be used in daily practice, it must be properly tested and calibrated. The accelerometer only provides numerical acceleration values, which need to be processed to be correctly interpreted and linked to specific behaviours intended for monitoring. Only after verifying the reliability of the processed data can the sensor be field-deployed.

One factor contributing to accelerometers’ success in animal behaviour monitoring may be their ability to detect both static tilt relative to the Earth’s gravitational vector and dynamic acceleration generated by the animal’s movements [17]. However, there are some limitations that can impact the accelerometer and the associated electronics’ usability. For example, when using a memory card, real-time behaviour information cannot be obtained, as data must be transferred to a computer for processing afterward. Additionally, the large amount of data generated by the accelerometer can quickly saturate the memory card, limiting the observation period. If the accelerometer is equipped with a transmitter, the animal can be remotely monitored in real time, but this may lead to rapid battery depletion, depending on the data transmission frequency.

Despite the many advantages offered by accelerometers, these sensors have certain intrinsic limitations that make them less suitable in specific circumstances. Firstly, during dynamic movements, the accelerometer detects both body segment tilt and dynamic acceleration due to movement, which can cause interference between the two parameters, making it challenging to separate them. In extreme cases, the accelerometer cannot be used to measure tilt, as the total detected acceleration approaches zero [17,43]. Additionally, for the same activity, signal intensity may vary significantly depending on the sensor’s position on the body [17,44], complicating the accurate estimation of behavioural parameters. Finally, accelerometers may not be ideal for detecting and characterizing dynamic behaviours involving slow movements, especially those based on rotation [17,43].

#### 5.1.1. Practical Applications

The indicators obtained from accelerometric devices, both commercial and experimental, pertain to the time animals spend on primary activities such as feeding behaviour (e.g., duration and frequency), ruminating (in both intensive and extensive systems), resting or inactivity, movement (walking, running, exploring), assumed pose (standing, lying down), and related transitions (when the animal stands up or lies down). These devices have been widely used to improve animal welfare, as highlighted by studies that have detected lameness in grazing dairy cattle [45,46] and beef cattle [45,47]. Sensor positioning should be determined according to the behaviour to be examined. A collar-mounted sensor seems suitable for predicting feeding behaviours (food intake, grazing, and rumination) and movement-related behaviours and restlessness [36]. In terms of feeding behaviour, research has focused on grazing activity, analyzing jaw movements to classify them as biting (grasping and tearing), chewing (crushing), and bite–chewing (overlapping chewing and biting activities), counting the number and duration of these movements to distinguish between grazing and rumination. Direct estimation of grass intake through accelerometers has been carried out using various methods, including collar-mounted devices that record daily activity balances, such as grazing [45,48], or with the assistance of bite counting [45,49]. However, caution is needed when using data from these sensors to estimate food intake to ensure nutritional requirements are met, considering that grazing can vary in composition and quality, and bite speed and mass differ depending on grass height, density, and dry matter concentration [45,50]. Compared to other devices like a GPS antenna, accelerometer has low energy consumption and is highly accurate in detecting head position, allowing for distinctions between grazing, lying down, and standing [45,51]. In some studies, accelerometers and GPS have been used in combination. Gou et al. (2019) compared three methods of classifying cattle activity in pastures and found that the tri-axial accelerometer model was the most accurate, achieving 96% accuracy in overall behaviour classification. In this paper, the authors calculated an accuracy of 98% for grazing behaviour, including foraging and walking–foraging behaviours; instead, they reported an accuracy of 92% for the nongrazing behaviours, where the latter include walking, standing, lying down, and rumination actions. This high accuracy value reflects the model’s ability to differentiate between grazing and nongrazing behaviours, primarily due to its reliance on instantaneous acceleration metrics [52]. However, location might be very important in grazing systems; hence, the GPS–tri-axial model or GPS-only model (90% accuracy) is more suitable for grazing dairy and beef cattle. Moreover, accelerometer technology, with its low power consumption, can extend GPS battery life by activating GPS recording only when the accelerometer detects movement at a certain speed [45,53].

Devices with integrated accelerometers for dairy and beef cattle are already available on the market, providing farmers with real-time monitoring of animal welfare by building historical activity trends at both individual and herd levels, alerting operators in the event of abnormal behaviours. Another interesting application of accelerometers, which are already available in various commercial devices, is the detection of hyperventilation and heat stress.

Few applications of accelerometers and RFID have been reported for studying drinking behaviour and water intake in grazing animals [45,54]. Water intake has been calculated using a trough equipped with a water flow meter. The combination of these technologies has enabled the number, duration, and frequency of each animal’s visits to a water point; the number and duration of drinking events per visit; and the time each animal spends drinking to be recorded [45,54]. Thus, further developments could allow farmers to monitor whether the herd’s water intake needs are met, even in challenging environmental situations such as dry seasons.

In recent years, the use of accelerometers and monitoring systems integrated with accelerometers has become popular for monitoring animal activity and predicting oestrus [45,55,56]. Beyond oestrus detection, accelerometers have also been used to detect calving times by monitoring variations in animal behaviour [57]. For instance, a study combining neck-mounted accelerometers on collars, leg-mounted accelerometers, and ultra-wideband (UWB) localization sensors found that this approach improved detection performance within 24–8 h before calving. The neck-mounted accelerometer on the collar achieved a Precision of 50–53% and Sensitivity of 47–48% (AUC 86–88%), while combining sensors further enhanced detection performance [57]. In some cases, parentage monitoring may also be relevant. An initial attempt in this area was made by Abell et al. (2017), using accelerometric data and various classification algorithms (random forest, random tree, and decision tree) to try to predict bull behavioural events in a multi-bull pasture [58]. The authors were able to distinguish between lying down, standing, walking, and mounting, but the accuracy for mounting events ranged from only 74% to 80%, which was insufficient [45].

Recently, the use of Inertial Measurement Unit (IMU) sensors has been reported. The IMU is a combined device that includes various sensors (accelerometer, gyroscope, magnetometer) capable of measuring linear acceleration, rotation angles (pitch, roll, and yaw), and angular velocity. An IMU from a standard mobile phone was used on cattle [29,45], achieving 92% accuracy in activity classification, reaching 95% for rumination activity.

#### 5.1.2. Existing Commercial Products

The collars identified on the market that currently support the integration of accelerometer sensors are: AfiCollar™ by Afimilk (Kibbutz Afikim, Israel), RealTime SmartTag^®^ by BouMatic (Madison, WI, USA), Cowlar^®^ by Cowlar (Memphis, TN, USA), MooMonitor+^®^ by Dairymaster (Cincinnati, OH, USA), DelPro™ by DeLaval Inc. (Tumba, Sweden), CowScout Neck^®^ by GEA Farm Technologies (Neumarkt In Steiermark, Austria), Qwes™ HR by Lely (Maassluis, The Netherlands), C-SENSE Cow collar by Milkline (Piacenza, Italy), SmartTag neck^®^ by Nedap (DC Groenlo, The Netherlands), Heatime™ by SCR Engineers Ltd., Allflex (Rahway, NJ, USA), HR-tag (SCR Engineers Ltd., Allflex), SenseHub™ Dairy^®^ (SCR Engineers Ltd., Allflex), and CowTRAQ™ by Waikato Milking Systems NZ (Hamilton, New Zealand). Table 1 collects the different studies adopting the specific products. In the reference studies associated with each commercial product are also reported the results of the validation of each product.

Shergaziev et al. (2024) examined the use of AfiCollar for collecting productivity, health, and reproductive data, confirming that the device provides detailed information for herd management and disease prevention by accurately measuring rumination and oestrus activity [59]. Leso et al. (2021) validated AfiCollar’s accuracy in monitoring rumination and feeding, finding a significant correlation between visual observations and sensor data [37]. The results confirmed the device’s suitability for various feeding scenarios, ensuring accurate data for both rumination and feeding times, with applications that can enhance dairy cow health management. Iqbal et al. (2023) used AfiCollar to analyze associations between grazing and rumination behaviours with performance parameters such as milk yield and milk composition in a grazing system [60]. The study showed that rumination time positively correlated with milk production and solid milk components, demonstrating the device’s utility in monitoring and optimizing grazing cow productivity.

Real-time, used in Kapusniaková et al. (2024) and already described in the section on commercial collars with integrated radio frequency identification, was employed to study differences in feeding and rumination times among cows in different lactation stages, revealing a negative correlation between dietary starch content and rumination time [61].

Verdon et al. (2018) used the MooMonitor to study rumination and rest behaviours in relation to feeding frequency on grazing strips, noting that more frequent feeding reduced rumination time and milk production [62]. Grinter et al. (2019) validated MooMonitor’s accuracy by comparing collected data with visual observation, confirming the device’s accuracy in studies on cow behaviour [63]. In Borghart et al. (2021), the MooMonitor was used to predict lameness in cows through behavioural and production data, developing a predictive model combining accelerometer data with production parameters [64]. Krpálková et al. (2022) explored the correlation between rumination time detected by the collar and milk production at different lactation stages, noting that cows with longer rumination tended to produce more milk [65]. Raedts et al. (2024) used MooMonitor to monitor feeding behaviour and milk production in grazing cows supplemented with concentrates, revealing a correlation between supplementation levels and milk response, with the device providing detailed data on intake and behaviour under grazing conditions [66]. Benaissa et al. (2023) integrated MooMonitor data with an ultra-wideband (UWB) localization system, demonstrating the use of a combination of accelerometer and localization data for more precise monitoring of cow health and behaviour [67]. Werner et al. (2019) validated the collar’s effectiveness in grazing behaviour monitoring, confirming the device’s high accuracy in measuring rumination and feeding time [68]. Moore et al. (2021) used the device to monitor oestrus in grazing cows, showing that the collar could promptly detect oestrus onset and optimize timing for artificial insemination, increasing pregnancy rates [69]. Finally, Beauchemin (2018) explored MooMonitor’s potential to monitor rumination and feeding activity, emphasizing the importance of these data in maintaining rumen health and early detection of feeding issues, such as acidosis [70].

Shergaziev et al. (2024) analyzed the effectiveness of DelPro in dairy farming, highlighting the software’s ability to track key parameters, including individual milk production, rumination, and reproductive activity [59]. With real-time monitoring and automatic report generation, DelPro optimizes health and fertility management, reducing decision times and improving herd operational efficiency.

The study by Nóbrega et al. (2019) explored the integration of collars with on-market sensors for location monitoring, including Digitanimal, Nofence, and eShepherd, and commercial collars with accelerometers for activity monitoring, such as Cowlar and Cowscout, within IoT systems for intelligent animal monitoring [71]. Madureira et al. (2020) also used CowScout to monitor oestrus behaviour and intensity, analyzing motor activity in relation to reproductive cycles [72]. The device was used to send real-time alerts about activity increases, indicating the optimal time for artificial insemination to improve conception rates and reduce reproduction-related costs.

Müschner-Siemens et al. (2020) used the Qwes collar to study how the daily rumination time in lactating cows decreased when a threshold temperature–humidity index of 52 was exceeded [73]. In another study, Gáspárdy et al. (2014) used Qwes to monitor variations in rumination, body weight, and milk production in cows with subclinical ketosis and uterine diseases [74]. The device detected significant reductions in rumination time days before the disease diagnosis, indicating its utility as an early warning tool for health issues before clinical signs appear.

Quddus et al. (2022) validated the use of SmartTag for monitoring feeding, rumination, resting, and standing behaviours, finding a high correlation between device data and video observations, suggesting the device is highly accurate for monitoring these activities in buffalo [75]. Dela Rue et al. (2020) tested SmartTag in a grazing context to measure feeding time, confirming that the device can effectively monitor feeding activity in grazing conditions, with feeding time data closely correlating with direct observation, making it a valuable tool for farms operating in grazing systems [76]. Zhou et al. (2023) used SmartTag to monitor oestrous-related motor and ruminal activity in dairy cows, showing that physical activity increased on the oestrus day compared to baseline [77]. Thomas et al. (2024) studied the repeatability and predictability of resting and feeding behaviours, discovering that SmartTag can accurately measure individual variations, aiding in early detection of behavioural changes related to stress or illness [78].

The Heatime collar system has been widely used to monitor oestrus in dairy cows. Borchardt et al. (2021) used the collar in the early postpartum period to detect oestrus [79]. Holman et al. (2011) evaluated its sensitivity and predictive value compared to other methods, noting limitations in high-producing cows with low body condition scores (BCS) [80]. Neves et al. (2012) compared reproductive efficiency between an artificial insemination program and an automated activity-monitoring system (AAM) with the Heatime collar, observing advantages in some herds with the AAM method [81]. Valenza et al. (2012) tested the collar’s effectiveness in detecting oestrus and the use of GnRH at insemination, with variable results [82]. Aungier et al. (2015) showed a correlation between oestrus activity detected by the collar and oestradiol levels, indicating the device’s ability to determine the optimal time for artificial insemination [83]. Silper et al. (2015) compared the Heatime collar with the IceTag device, finding both to be accurate but with differences in detection intensity [84]. Madureira et al. (2015) highlighted that BCS and parity affect the intensity and duration of oestrus activity, showing that the system is useful for monitoring higher-parity cows [85]. Burnett et al. (2017) associated a longer oestrus duration monitored by the collar with higher pregnancy rates, facilitating insemination planning [86]. Moretti et al. (2017) monitored the impact of heat stress through the collar, showing a negative correlation between RT and the temperature–humidity index [87]. Veronese et al. (2019) combined Heatime with PGF2α treatments to reduce interservice intervals [88]. Plenio et al. (2021) developed BovHEAT, an open-source tool for automatic data analysis from the collar [89]. Reed et al. (2022) studied oestrus activity in relation to genetic fertility [90]. Macmillan et al. (2022) used the collar to predict calving based on activity and rumination changes, noting increased activity 8 h before calving and a decrease in rumination [91]. Vincze et al. (2024) employed Heatime to monitor cavitary corpora lutea, observing that their presence significantly reduced the chances of conception [92]. Tippenhauer et al. (2023) monitored oestrus activity intensity using Heatime and SmartTag neck collars, finding that high activity intensity correlated with higher pregnancy rates in cows undergoing timed insemination [93].

The HR-Tag collar was used by Giaretta et al. (2019) to monitor rumination time and activity, observing how these parameters varied in response to diet, environment, and animal health [94]. In another study, Teixeira et al. (2022) used the HR-Tag and Heatime collar to detect early anaplasmosis in dairy calves, demonstrating how changes in RT and AT could predict disease days before clinical signs appeared [95]. By analyzing data using predictive models, the study highlighted the effectiveness of the two collars in early disease detection, improving health management. Finally, Lemal et al. (2024) studied SenseHub’s effectiveness in monitoring key behaviours such as activity time (ACT), rumination time (RUM), and feeding time (EAT), particularly in relation to heat tolerance [96]. The results showed that SenseHub could detect significant behavioural changes in response to heat stress, with a positive correlation between activity and fat- and protein-corrected milk production (FPCM) and a negative correlation with somatic cell score (SCS). Additionally, the device allowed data collection from non-lactating cows, expanding genetic evaluation possibilities for heat tolerance.

### 5.2. Gyroscope

This device bears some similarity to accelerometers, but instead of measuring acceleration along an axis, it measures angular velocity and is thus used to obtain information about the rotation of a body. Currently, it is rarely used, except in a few experimental cases in conjunction with accelerometric devices, to improve the accuracy of movement measurement [36,97,98]. The gyroscope sensor can simultaneously detect angular movements along three mutually perpendicular axes [99].

#### 5.2.1. Practical Applications

The primary function of the gyroscope sensor is to detect and record the angular movements of cattle, which are essential data for analyzing their motor behaviour during the oestrous cycle [99]. To extract sufficient insights, in some behaviour recognition systems, gyroscope data are also used as a complement to accelerometer data [97,98,100]. When using a combination of accelerometer and gyroscope sensors, it is possible to distinguish between different physical activities with similar movement data, achieving a higher recognition rate compared to when the sensors are used separately [100].

#### 5.2.2. Existing Commercial Products

From the research conducted, no commercial collar devices for dairy cows have been identified that explicitly incorporate gyroscope technology. This could suggest that, while gyroscopes are widely used in scientific research studies for detailed analysis of angular movements and posture, their integration into commercial collar solutions remains limited. The absence of gyroscope-equipped collars on the market may be attributed to several factors, including technical challenges related to sensor calibration and data interpretation in real-world farm conditions, as well as cost constraints associated with integrating this technology into commercially viable products. Further research should explore how gyroscopes could enhance collar-borne sensing systems for dairy cattle, improving the detection of behaviours such as grazing, feeding, and rumination. Investigating these technologies in combination with accelerometers and GPS could pave the way for a more comprehensive understanding of cattle activities and welfare, contributing to more advanced and effective solutions for PLF. Additionally, it is possible that some commercial devices may already include gyroscopic components without explicitly highlighting them in their specifications. Greater transparency in device specifications would enable researchers and farmers to better assess the potential of existing technologies.

### 5.3. Magnetometer

Magnetometers share many similarities with accelerometers. They are sensors capable of measuring both a static component, through inclination relative to the Earth’s magnetic field, and a dynamic component, which corresponds to changes in the sensor’s inclination over time [17,43]. Data recorded by this device are generally used in practice as a complement to accelerometer data in certain behaviour recognition systems, with the aim of extracting sufficient insights [97,98]. Despite the similarities between accelerometers and magnetometers, the latter do not experience the issues identified with accelerometers. First, the magnetometer measures the sensor’s inclination. The dynamic component is not mixed with the static component of the signal and can be obtained by differentiating the signal with respect to time [17,101]. However, it is worth noting that when the axis of rotation aligns exactly with the direction of the local magnetic field—although this is a rare occurrence—the dynamic component will be zero [17,43]. The equivalent operation for accelerometers, i.e., integration over time, does not directly provide velocity due to the need to resolve the integration constant using knowledge of the initial or final velocity from another source. Secondly, it has been shown that magnetometers are capable of distinguishing behaviours that are not easily identifiable using accelerometers alone [17,43].

#### 5.3.1. Practical Applications

The static component of the magnetometer is widely used to obtain the animal’s orientation, while the dynamic component is generally used to extract metrics that describe angular velocity [17,101]. In the context of animal behaviour recognition, tri-axial accelerometers and magnetometers are employed to identify movement patterns in animals. Although accelerometers have been used much more extensively, recent studies have shown that magnetometers can more accurately describe certain low-acceleration behaviours. In fact, a recent comparison between accelerometers and magnetometers demonstrated that significant quantifiable differences could arise in the recognition capability of the two sensors for specific behaviours [17,43].

#### 5.3.2. Existing Commercial Products

Based on the research conducted thus far, no commercially available collar devices for dairy cows have been identified that explicitly incorporate magnetometer technology. While magnetometers are frequently employed in scientific research studies to assess head orientation and angular movements with high precision, their integration into commercial collar solutions appears to be limited.

This limitation may stem from technical challenges related to sensor calibration and the susceptibility of magnetometers to environmental interferences, such as metal structures commonly found in farm facilities, which could compromise data accuracy. Additionally, the complexity of processing and interpreting magnetometer data in real-world farm conditions may pose further barriers to their adoption in commercial products.

Future research should focus on overcoming these challenges and exploring how magnetometers could be effectively integrated into collar-based systems. The inclusion of magnetometers might enhance the precision of behavioural monitoring by providing accurate measurements of head orientation and movements, which are critical for detecting behaviours such as grazing and feeding. Moreover, combining magnetometers with accelerometers and gyroscopes could enable a more comprehensive understanding of cattle activities and welfare, potentially offering new avenues for innovation in precision livestock farming technologies. It is also possible that some commercial devices may include magnetometer components but do not explicitly highlight their presence in product specifications. Greater transparency from manufacturers would help researchers and end-users evaluate the full potential of existing technologies.

### 5.4. Microphone

Among the various tools that can be used in practice, acoustic sensors can also be programmed and employed on farms to monitor animals and detect signs of disease or other health and welfare issues. Vocalizations are recorded continuously and, via a transmitter, sent to a fixed receiver connected to the computer’s audio card, where the data are transformed and analyzed using specific audio-editing software. Using an available algorithm, sound recordings are segmented into serial signal windows, and only those that exceed a defined threshold are considered for the detection and classification of specific behaviours. Information is encoded in a cow vocalization event through its duration, frequency, type (closed or open mouth), and dynamic changes in its spectral and power components that can be represented by a spectrogram. By analyzing the main features of the spectrogram over time, it is possible to evaluate the time series of the behaviours of the monitored animal [102].

Although currently used primarily for scientific purposes, this technology shows promising potential for the future [36]. The sound sensor is based on a condenser microphone integrated with a signal amplifier (LMV324) which, together, are used to detect and record cattle vocalizations. This sensor is designed to capture the number of vocalizations emitted by livestock [99]. A potential advantage of using these devices is the ability to record surrounding environmental sounds as well. This multi-component approach could provide a more comprehensive picture of farm conditions and help identify potential stressors linked to the environment. Additionally, microphones can be used to further study interactions between animals, allowing for a better understanding of social behaviour. Despite these advantages, the complexity of multi-component analysis poses a challenge, as it requires distinguishing between sounds produced by individual animals in environments where vocalizations frequently overlap. This may necessitate a network of synchronized microphones among multiple animals, increasing the difficulty. Furthermore, durability issues may arise in extreme climatic conditions. Environments characterized by heavy dust, humidity, or fluctuating temperatures can compromise performance. In such contexts, microphones should be equipped with advanced protection or self-heating technologies, which leads to additional implementation and maintenance costs.

#### 5.4.1. Practical Applications

The data recorded by the microphone can provide information about the sounds produced by animals, including their frequency, intensity, amplitude, and duration. These characteristics are correlated with the animals’ emotional state and can thus offer important insights into evaluating their emotional state in terms of valence, which can be positive or negative, and arousal, corresponding to the level of activation of the nervous system, which can be relatively high or low. This information is useful for identifying situations of discomfort or, conversely, positive emotional states [103]. The vocal behaviour of cattle can also provide information about the reproductive state of the vocalizing animal; during the oestrus period, for example, the vocalization rate is increased [104]. Sound analysis also allows the identification of pathological sounds, which can be common signs of respiratory diseases, such as coughing and sneezing [36].

Acoustic sensors can also be used for monitoring feeding behaviour, providing an estimate of food intake in pastures [45,105,106,107,108,109]. They primarily rely on a microphone positioned on the animal’s head or near the mouth [110]. The acoustic signal is recorded, and the frequency, intensity, duration, and time between events are used to classify them as biting and chewing events. Signal classification is performed subsequently. To extend monitoring and reduce the required storage space, systems with an integrated processor have been developed to run algorithms for the automatic and real-time classification of acoustic signals into chewing, biting, and bite–chewing events across various livestock species [45,111,112].

#### 5.4.2. Existing Commercial Products

The collar devices currently on the market that integrate sensors capable of recording sounds are Qwes™ HR (Lely) and Hi-Tag (SCR Engineers Ltd.).

As previously mentioned in sections regarding other collar sensors, Gáspárdy et al. (2014) used the Qwes collar to analyze how rumination can serve as an early indicator of subclinical diseases, showing that variations in rumination times can predict conditions such as subclinical hyperketonaemia [74]. Müschner-Siemens et al. (2020) studied the impact of heat stress on rumination activity using the Qwes system, highlighting a decrease in rumination as the temperature–humidity index increased, confirming the system’s effectiveness in monitoring resilience to environmental stress [73].

Several studies have validated the effectiveness of the Hi-Tag collar for monitoring rumination, demonstrating its usefulness in both confined environments and grazing animals. Schirmann et al. (2009) initially validated this system applied to dairy cows, finding a significant correlation between the data recorded by the collar and direct observations [113]. Burfeind et al. (2011) expanded the research by applying the system to young animals, verifying that Hi-Tag is also accurate for monitored rumination in heifers of at least 9 months old [114]. In fact, for 9-month-old heifers, rumination times recorded by the independent observers were highly correlated with Hi-Tag outcomes (R^2^ = 0.77) and the mean difference was about 4 min out of a monitored period of 2 h (equivalent to a percentage difference of 8% between observer and device output data). Andreen et al. (2020) correlated the rumination time recorded by the collar with milk fat production, suggesting that Hi-Tag can detect variations in ruminal efficiency and productive parameters [115]. Vanrell et al. (2018) applied the system in grazing conditions, developing an algorithm to improve the distinction between feeding and rumination behaviours using Hi-Tag’s acoustic signals [116]. Held-Montaldo et al. (2021) used the collar to investigate how metritis and weather conditions influence lying and rumination behaviours in transition dairy cows under seasonal pasture-based systems [117]. Recently, Mirzaei et al. (2023) confirmed Hi-Tag’s ability to monitor individual variability during the peri-partum period, observing the device’s effectiveness in detecting behavioural changes related to health status [118].

### 5.5. Radio Frequency Identification Tags

Radio frequency identification (RFID) tags, which can also be applied in a collar-based system, currently represent the most used technique for identification in cattle farming [73]. This approach uses low-frequency (LF) radio waves (with an animal identification frequency of 134.2 kHz), but high-frequency (HF) and ultra-high frequency (UHF) systems are also available [36]. This technology allows for simple and accessible identification, tracking, and monitoring of livestock, enhancing traceability throughout the supply chain [45,119]. At the practical farm management level, it has enabled the development of management software that automatically stores individual daily records, including information such as growth, medical treatments, pedigree, and productive or reproductive performance [120,121]. Structurally, the system consists of an Electronic Identification Device, known as a transmitter, which comes in the form of a tag (a transponder or tag), and a receiver that acts as a signal reader. The components are lightweight, robust, reliable, and durable, provided the transponder’s electronics are protected. Animal identification can be performed manually by an operator or fully automated by positioning transponders in strategic, mandatory passage points of the structure, such as near the milking parlour. This technology not only enables automatic sorting of animals through gates for routine tasks, such as insemination, but also offers the advantage of remote operation, as no direct visual contact is required between the transponder and the responder [36]. One major challenge of RFID technology is the tag’s battery life, which in some systems has been addressed by drawing power directly from the antenna. Concerning readability, there are mixed results; in crowded conditions, such as in a milking parlour, reading failures may occur [122], while in static conditions in which animals are restrained, a 100% readability rate has been reported using a handheld tag reader [123].

#### 5.5.1. Practical Applications

As previously mentioned, farms equipped with automatic sorting gates can divide animals within the herd based on specific characteristics of interest using RFID technology. Animals that have reached slaughter weight, females close to calving, newborn calves, or those requiring dietary supplementation or specific medical treatments can be directed to designated areas [124,125]. Additionally, some systems for reconstructing maternal pedigree rely on RFID technology. Herds or pastured groups often consist of fertile females and males, where the main challenge is to establish the pedigree of the offspring [38]. By recording the sequence in which RFID tags are read as animals pass near the reader when entering an enclosed area containing an attractant, it becomes possible to assign each calf to its mother. Although it required 21 days of data collection to achieve 80–85% maternal linkage in a herd of 41 beef cattle [126], this system is considered less labour-intensive and less costly than manual capture or DNA comparison, and it can also improve genetic progress in pasture-based breeding systems [45].

#### 5.5.2. Existing Commercial Products

Based on the research conducted, the main collar devices on the market currently featuring an integrated RFID system are as follows: RealTime SmartTag^®^ (BouMatic), Tru-test Active Collar Tag (Datamars), Qwes™ HR (Lely), C-SENSE Cow collar (Milkline), and HR-tag (SCR Engineers, Allflex). These commercial solutions have been employed in various scientific studies. In Kapusniaková et al. (2024), dairy cows were equipped with RealTime SmartTag^®^ collars (BouMatic), which integrate a transponder for individual identification [61]. The collected data were downloaded every 24 h through a digital receiver located in the barn, using the RealTime Activity 3.0 software; this system enabled detailed and continuous monitoring of the cows’ behavioural activities, providing crucial information on rumination time and feeding. A study by Williams et al. (2019) assessed the effectiveness of the Tru-test Active Collar Tag (Datamars) in monitoring cattle visits to three different watering points in northern Australia [127]. The collected data showed that most water point visits occurred during daytime hours, with a significant increase in conditions with a higher temperature–humidity index and greater cloud cover. The information obtained via RFID allowed the identification of behavioural differences in drinking frequencies, correlated with variables such as climate, water availability, and seasons. The Qwes™ HR rumination monitoring collar (Lely) was adopted in the study by Müschner-Siemens et al. (2020); it enabled individual animal identification during automatic milking (AMS), allowing accurate collection of rumination data [73]. This same collar was also used in the study conducted by Gáspárdy et al. (2014), allowing the collection of data on daily rumination activity, live weight, and daily milk yield [74]. Identification was essential for tracking each animal during the automatic milking phase, providing a detailed picture of behavioural and productive parameters. Giaretta et al. (2019) used the automatic HR-tag collar (SCR Engineers, Allflex) to record rumination time and activity [94]. Data, transmitted every 20 min to software via an antenna, allowed continuous monitoring of animal behaviour and analysis of daily patterns. Similarly, Lamanna et al. (2024) utilized this collar to identify animals at the milking robot and evaluate their behaviour, focusing on habituation during robotic milking sessions [113].

### 5.6. Global Positioning System Receivers

The GPS is used for identification and positioning/localization, particularly in extensive farming conditions, where enclosing animals with fences is both costly and considered an ecological constraint; fences are often unwelcome as they impede the free movement of wildlife. This technology can also be applied indoors by using pseudolites, which replace the satellite system. The system comprises radio collars worn by the animals and a set of satellites in orbit. The radio collars’ signals are received by the satellite system at variable frequencies. By processing the three geocentric coordinates (longitude, latitude, altitude), the exact position of each animal is determined, along with additional information regarding the distance travelled over a given period [36]. Despite the potential practical applications of GPS for monitoring grazing livestock, a significant obstacle to its widespread adoption on farms is the high cost. Equipping each animal with a GPS device often imposes a prohibitive economic burden on most farmers. As a result, efforts have been made to reduce the cost of each device. For example, Karl and Sprinkle (2019) experimented with low-cost collars, priced at EUR 51, using commercially available electronic components [115]. However, despite their affordability and ease of use, these collars had certain limitations compared to commercial GPS devices, such as a battery life limited to a few weeks and a lack of wireless data transmission [128,129]. To address issues related to wireless transmission and the high costs of GPS tracking solutions, Maroto-Molina et al. (2019) developed and tested a low-cost solution in a real-world setting [130]. This solution involved equipping some animals in the herd with GPS collars connected to a Sigfox network, while the remaining animals wore low-cost Bluetooth tags. Another limiting factor for the adoption of GPS devices for livestock monitoring is battery life. In grazing contexts, livestock management is minimal, and manual interventions are infrequent and spaced out. For tracking systems to be effective, they must cover the entire grazing season, minimizing or eliminating the need to replace batteries. This issue has been partially addressed by integrating solar panels into the devices. A network architecture designed for herd monitoring, where most nodes are powered by the animals’ movement, has been successfully tested to track reindeer herds in Lapland [45,131]. Furthermore, GPS locators have been combined with other tools, such as accelerometers and temperature sensors, to monitor animals’ physical activity and health status. In relation to the application in intensive systems using pseudolites, this solution is still rarely used due to its costs and because signals can be distorted by obstacles [36].

#### 5.6.1. Practical Applications

In addition to the spatial distribution of animals and the most frequented grazing sites, data recorded with GPS can provide valuable information for identifying and classifying a wide variety of activities carried out by the animals [45,132], including changes in walking gait, resting or inactivity behaviour, feeding and rumination (meal frequency, drinking, etc.), and distribution in relation to climate [133]. These are all important indicators of changes in animal welfare, which has become a priority in recent years [134]. Systems comprising GPS trackers and aerial monitoring of pastures have been tested as support tools for grazing planning, helping to prevent overgrazing and grassland degradation. A cloud-based and WebGIS grazing management system has also been proposed, which is capable of displaying the herd’s real-time location, historical trajectory, and monitoring and estimating forage growth and intake through both UAV and satellite RS images. This information can be accessed by end-users in real time, providing a decision-making basis for herd management.

A GPS-integrated collar can also detect calving events in grazing systems [135]. For this purpose, a GPS-based calving alert device has been patented, which notifies the farmer via SMS, reporting the date and time of the calving event, the animal’s ID, and the geographic coordinates of the calving location [135]. The GPS coordinates are imported into a common mobile application; the device can cover up to 10 births per year at a unit cost of EUR 31.5 per birth. Additionally, the spatiotemporal data from GPS devices, with appropriate mathematical modelling, could be used to identify predation and alert the farmer. Sendra et al. (2013) proposed a prototype of an intelligent wireless sensor network comprising a system that measures heart rate and body temperature, with data interpreted by an intelligent algorithm capable of detecting collective stress episodes caused by predator attacks during the night [136]. When an attack occurs, the system automatically triggers acoustic and visual alarms to scare off predators and sends an alert signal to the farmer. This prototype should be tested under farm conditions but offers some useful features for further implementation, such as recharging via a solar panel and a control unit that limits its operation to nighttime, considering energy constraints in field conditions. This functionality could have a real impact in preventing predation, as it can provide an immediate response to scare off predators while awaiting human intervention [45].

Another emerging use of GPS is virtual fencing (VF). Through this system, farmers can select and delineate an area where livestock can graze, thanks to the integration of the global mobile communication system (GSM). The traditional physical boundary is replaced by an acoustic stimulus; when the animal approaches the virtual fence, an audio signal warns it to stop. If the animal ignores the signal, it receives an electric shock. The system consists of collars with a GPS locator and a battery-powered device that administers the electric shock [36,45,137].

#### 5.6.2. Existing Commercial Products

Currently, the main collar devices on the market that integrated GPS technology are as follows: Digitanimal^®^ by Digitanimal (Madrid, Spain), eShepherd^®^ by Gallagher (West Branch, MI, USA), Halter^®^ by Halter USA Inc. (Auckland, New Zealand), PinnaclePro Series by Lotek Engineering Inc. (Newmarket, ON, Canada), Vence^®^ by Merck & Co., Inc. (Rahway, NJ, USA), and Nofence^®^ by Nofence (Molde, Norway). These devices have been evaluated in various scientific studies, highlighting both their advantages and limitations.

Digitanimal^®^ was examined in studies by Hassan-Vásquez et al. (2022) and Nóbrega et al. (2019) [71,138]. The first study demonstrated the collar’s utility in real-time tracking of livestock movements in the dehesa, a typical agroforestry system of the Iberian Peninsula, monitoring environmental impact, such as manure distribution. The GPS collar transmits data at regular intervals, as often as every 30 min, and operates with a long-lasting battery and SigFox network support. However, limitations in data collection frequency and network coverage affect monitoring accuracy in areas with challenging environmental conditions. The second study delved into the integration of collars with GPS sensors available on the market, including Digitanimal, Nofence, and eShepherd, and commercial collars equipped with accelerometers for activity monitoring, such as Cowlar and Cowscout, within IoT systems for intelligent animal monitoring. The aim was to examine the potential of these devices in providing detailed information on animal health and welfare [71,138].

The features of the PinnaclePro collar were discussed in a study conducted by Sigüín et al. (2022), which proposed a modular collar for animal telemetry, designed with an animal-centred approach to address the limitations of different commercially available devices [139]. Among these commercial devices, some have been noted for their application in virtual fencing. Nofence^®^ emits an acoustic signal when the animal approaches the virtual boundary and provides a mild electric pulse if the limit is crossed. A distinctive feature of this system is the “teach mode”, which facilitates learning in the first few days by deactivating the electric pulse more easily, allowing the animal to associate the sound with the boundary. Simonsen et al. (2024) analyzed the effectiveness of this collar in helping cattle adapt, highlighting that the process is particularly advantageous when animals are introduced to the system in the presence of other already “experienced” animals [140]. Hamidi et al. (2024), using the same collar, observed that after about 12 days, cattle learn to associate the acoustic signal with virtual boundaries, reducing the need for electric pulses [141]. Lund et al. (2024) explored the possibility of placing the collar only on leader animals, hypothesizing that the group would follow them. However, no predictable pattern was found in group behaviour, suggesting that effective control requires collars on each animal, especially for large herds and complex terrains [142]. Goliński et al. (2023) conducted a comparison between the various collar systems, including Nofence, Vence, Halter, and eShepherd, highlighting the advantages and limitations of each and showing how each technology meets specific needs, offering farmers various options to improve livestock management in terms of sustainability, animal welfare, and ease of use [143].

## 6. Smart Collars on the Market

As the adoption of PLF technologies grows, commercial solutions for smart collars have become integral to modern cattle management (see Table 1).

**Table 1 animals-15-00458-t001:** Summary of commercial collars for dairy cattle monitoring presented alphabetically by the manufacturer’s name; within each manufacturer, they are listed alphabetically by their commercial name.

Commercial Name	Sensors	Manufacturer	Related Studies
AfiCollar™	Accelerometer	Afimilk	[37,59,60]
RealTime SmartTag^®^	Accelerometer and RFID	BouMatic	[61]
Cowlar^®^	Accelerometer	Cowlar	[71]
MooMonitor+^®^	Accelerometer	Dairymaster	[62,63,64,65,66,67,68,69,70]
Tru-test Active Collar Tag	RFID	Datamars	[127]
DelPro™	Accelerometer	DeLaval Inc.	[59]
Digitanimal^®^	GPS	Digitanimal	[71,138]
eSheperd^®^	GPS	Gallagher	[71,143]
CowScout Neck^®^	Accelerometer	GEA Farm Technologies	[71]
Halter^®^	GPS	Halter USA Inc.	[143]
Qwes™ HR	Microphone, Accelerometer and RFID	Lely	[73,74]
PinnaclePro Series	GPS	Lotek Engineering Inc.	[139]
Vence^®^	GPS	Merck & Co., Inc.	[143]
C-SENSE Cow collar	Accelerometer and RFID	Milkline	
SmartTag neck^®^	Accelerometer	Nedap	[75,76,77,78,93]
Nofence^®^	GPS	Nofence	[71,140,141,142,143]
Heatime™	Accelerometer	SCR Engineers Ltd. (Allflex)	[79,80,81,82,83,84,85,86,87,88,89,90,91,92,93,144,145,146]
Hi-Tag	Microphone	SCR Engineers Ltd. (Allflex)	[113,114,115,116,117,118]
HR-tag	Accelerometer and RFID	SCR Engineers Ltd. (Allflex)	[94,95,128]
SenseHub™ Dairy^®^	Accelerometer	SCR Engineers Ltd. (Allflex)	[96]
CowTRAQ™	Accelerometer	Waikato Milking Systems NZ	

Despite the numerous advantages offered by PLF, the collar-integrated sensors available on the market still present important limitations. Firstly, they are currently unable to estimate the kilograms of feed ingested and the time spent at the feed bunk by individual animals. The required precision for feeding monitoring tools (as well as the interest in validation) might increase if the data from these tools were integrated into marketing or health monitoring systems [10,147]. Additionally, a significant limitation in the development of behavioural classification models lies in their ability to classify multiple behaviours simultaneously with high accuracy. Certain behaviours are inherently more critical for specific farming objectives, such as early disease detection or reproductive management, while others may be less relevant. This variability underscores the need for tools that allow customization based on individual farms’ priorities. Farmers should have the flexibility to select models or devices tailored to their business needs, enabling them to focus on the behaviours that are most pertinent to their operational goals.

Furthermore, many commercial devices are designed to monitor only a limited number of parameters, which necessitates the use of multiple devices to obtain a complete overview of the animal’s status, implying a costly burden for the farmer and compromising practicality. Despite PLF technologies offering significant advancements, their increasing reliance on IoT-based devices raises important concerns about data privacy and security. Farmers often worry about unauthorized access, ambiguous data-sharing agreements, and the potential misuse of sensitive farm data by agricultural technology providers. These challenges can lead to hesitation in adopting new technologies, undermining their potential benefits. Addressing these concerns requires a collective effort among stakeholders—farmers, technology providers, and policymakers—to establish transparent data governance frameworks, promote digital literacy, and implement robust security measures, such as encryption and anonymization of data, to safeguard farmers’ and farms’ information [148].

Another key challenge is the cost-effectiveness of these systems and their return on investment. The substantial upfront costs of acquiring and implementing collar-integrated sensors, combined with ongoing maintenance and update expenses, can discourage farmers, particularly those managing smaller-scale operations. Additionally, the lack of clear economic validation or concrete evidence of long-term profitability further complicates adoption decisions. Addressing these economic considerations, alongside advancing technological solutions, should be a primary focus of future research to promote wider accessibility and adoption of PLF technologies [149].

It is also important to recognize that the success of these technologies depends not only on their capabilities but also on their usability and the readiness of farmers to integrate them into their operations. Managing the volume and complexity of data generated by these systems can be challenging, especially for farmers who may lack the necessary technical skills or resources. Ensuring that these technologies are user-friendly and accompanied by adequate training and support is essential to maximize their impact and adoption. Social media is emerging as a growing platform for both information and education in the field, providing an opportunity for farmers to exchange experiences, receive expert advice, and keep abreast of the latest advancements in PLF [150].

Finally, one of the key issues is the high energy consumption of the sensors, which require frequent recharges or battery replacements. This interrupts continuous monitoring and increases operational costs, especially in large farms. Continuous technological evolution can play a crucial role in resolving the energy duration issue of the sensors. At the same time, the development of more advanced sensors and devices integrating a broader range of sensors capable of monitoring multiple parameters simultaneously would reduce the need for multiple devices, simplifying farm management and improving efficiency.

## 7. Research Limitations

This study employed a structured literature review approach, but some limitations must be acknowledged. The reliance on peer-reviewed journal articles from Scopus, Web of Science, and Google Scholar may have excluded insights from grey literature or non-indexed sources. Despite using multiple keywords and synonyms with Boolean operators, some relevant studies might have been unintentionally missed due to database indexing differences. Additionally, the snowball sampling method, while useful for identifying key references, introduces an element of subjectivity. Furthermore, although Web of Science was searched broadly, specific sub-databases such as SCIE and SSCI were not exclusively targeted. Lastly, the study’s focus on collar-mounted sensors intentionally excludes insights from other sensor attachment sites. Despite these limitations, the methodology aimed to ensure comprehensive coverage and relevance in capturing the current state of knowledge on collar-mounted sensor technologies.

## 8. Future Perspective and Conclusions

Smart collars for PLF have quickly become indispensable tools in the livestock industry, revolutionizing how farmers manage cattle health, behaviour, and productivity. These devices, which are equipped with advanced sensors, provide unparalleled insights into the daily activities and well-being of individual animals. They enable early detection of subtle behavioural changes, allow for timely identification of health issues, and optimize essential processes like feeding and reproduction with minimal manual intervention.

The future of these systems lies in the development of more advanced, energy-efficient designs that incorporate multiple sensors capable of capturing a broader range of parameters. These technologies show great promise in enhancing data accuracy and enabling more precise monitoring of animal behaviours.

However, key challenges remain, particularly in terms of sensor integration, data processing, and usability for farmers. Addressing these challenges will be crucial for the next generation of devices, which are expected to deliver even more comprehensive data, further enhancing the efficiency of farm management. Continuous innovation and further research into sensor fusion, data analytics, and user-friendly systems will be critical for ensuring the sustainable and responsible use of PLF technologies in livestock farming.

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
