# Peer review of "Wearable Collar Technologies for Dairy Cows: A Systematized Review of the Current Applications and Future Innovations in Precision Livestock Farming"

_animals, 2025, doi:10.3390/ani15030458_

Round 1

Reviewer 1 Report

Comments and Suggestions for Authors

Overall

-       It is unclear as to why the review focuses only on collar-mounted sensors. It needs to be made absolutely clear the justification to only focus on collars and this needs to coincide with the contribution of this review to the current literature. There are dozens of reviews available now in similar areas, so this is an important aspect to get right.

-       It is sometimes unclear if the discussion is about collar-borne sensors or others.

-       There is overreliance on Aquilani et al., (2022).

-       There is overreliance on Abeni et al., (2023)

Section 1.

Line 44: delete ‘if’.

Line 53: It is not clear which context you mean here. Please be specific.

Section 2.

Line 55: Not all technologies are non-invasive. Please remove or be more specific.

Line 67: I suggest a full stop after ‘environment’…..This supports the decision making process…

Furthermore, please provide examples of instances where a farmer would perform the action and where the action would be fully automated.

Line 70: Again, please provide some examples of manual interventions that are reduced given these technologies.

For the countries listed you need to provide examples of the types of technologies that are being developed there and discuss the way that they are utilised. I don’t think it is enough to say that some countries produce more publications on PLF, and these are the reasons why. It is unclear why you say that the propensity for innovation is greater in Italy – why? Please state what the ‘different approaches  and priorities’ are in China.

Section 3

102-103: Please provide some examples of sensors that meet this statement. Also, it is unclear whether you’re discussing commercially available sensors or those used in research. I note you state ‘dataloggers’ in the next sentence which store data on-board, so do you mean a research context?

Line 117: In what context are these ‘most commonly used sensors’ used?

Line 117: What do you mean by the following: activity sensors, heat detectors, environmental sensors (provide some examples).

Line 124: Forehead sensors? Please provide an example.

Section 3 (end): We need to know exactly what the objectives of this study are and what this paper brings to the research area that we do not know about already. Furthermore, we need to know why collar-mounted sensors were selected and other not included?

Section 4

Line 139-140: What do you mean by ‘enhancing sensor integration in collar-based systems’.

Line 142: For what purpose was the review undertaken? I believe this will make more sense after you add clear objective at the end of the Introduction.

Line 142: It is unclear to me which review type you refer to in the Grant and Booth (2009) paper.

148-150: We need more information on how these terms were used and whether these were the only terms used. I imagine many more were used given the various sensors that are listed in Section 3. 

Line 152-161: Again, more information needed on the ‘snowball’ method mentioned. The description is currently limited and does not provide sufficient detail to understand how papers were selected.

Line 163-164: Why was the focus on collar-mounted sensors only? Please make clear why this was the case.

Section 5

Line 212: I would argue that it is the success of the data analyst not the accelerometer. It would be better if you talked about the benefit of the instrument directly.

Line 244: Do you mean feeding time? I’m not sure whether collar-based systems are adequate for feed intake.

Line 262: What does this measure relate (96%) to? Please be clear of the breakdown of performance.

Lines 267-271: Are these collar-mounted devices that you refer to dairy and beef cattle? If so, please provide some examples.

Line 275: Please define ‘well-identified’. We need to know the performance of these models. Furthermore, performance is also reported comparable to sensors mounted in ear tags.

Line 284: Typo – oestrus/estrus.

Line 286: Abeni et al., (2022) is unavailable to read. Please attach this paper with your responses so that it can be assessed. Otherwise please provide a different reference to the use of collar-borne accelerometers for predicting calving.

Line 299: “Existing commercial products” is better in my opinion because you are not defining a problem that needs a solution.

Section 5.1.2

I think much of what is said in this section could be put into a table with a much more succinct description of what each research paper described. Furthermore, I would expect the performance or the measured parameters to be included in this.

More importantly however, much of what is said in this section is similar to what is already provided in other publications such as this one Stygar et al (2021). It comes back to the original point; what does this paper provide that is s different to what currently exists?

Stygar, A.H., Gómez, Y., Berteselli, G.V., Dalla Costa, E., Canali, E., Niemi, J.K., Llonch, P. and Pastell, M., 2021. A systematic review on commercially available and validated sensor technologies for welfare assessment of dairy cattle. Frontiers in Veterinary Science, 8, p.634338.

Section 5.2.1

Much more information on performance needed rather than a plain narrative.

Section 5.2.2

More needs to be said here about where the research should go and why. What more could magnetometers bring to collar-borne sensing of dairy cattle?

Section 5.3.2

More needs to be said here about where the research should go and why. What more could magnetometers bring to collar-borne sensing of dairy cattle?

Section 5.4

More is needed in this section on the research undertaken with cattle using acoustic sensors. It would also be good to have more on the practicalities of this type of system mounted on a collar.

Line 490: I would also say that this was a disadvantage in the management of individual animals.

Line 513: Please state how likely this is to be better than conventional methods of heat detection such as activity recognition.

Line 532: What were the recorded behaviours?

Line 534: In what respect were they accurate for heifer?

Section 5.5

Please make very clear in this section how it adds to the existing literature. There are many other papers available that discuss the use of RFID.

Table 1. The title is too long. An extensive description of what is in the table is not required.

Section 6.

Lines 758-759: But there are limitations in model development with respect to classifying multiple behaviours as the literature shows. I would expect to see a discussion about this here. There are some behaviours that are more important than others and farmers should be able to select a tool based on their business needs.

I also expect to see other sensors discussed here such as those mounted in ear tags and the benefits of these over collars.

Line 768: We cannot overlook that farmers may not want or know how to deal with all this information either. This needs to be discussed here.

Reviewer 2 Report

Comments and Suggestions for Authors

Comment 1:

The article details a variety of sensor technologies, such as accelerometers, RFID tags, and GPS receivers, which are used in monitoring cow behavior, health status, and productivity. It is suggested that the authors further discuss how these technologies can be integrated into existing agricultural management systems and how they can work in concert with existing technologies such as automated milking systems.

To enhance the depth and breadth of the discussion on precision livestock farming and smart farming technologies within the manuscript, the following references are recommended for inclusion:

Mengjie Zhang, Yanfei Zhu, Jiabao Wu, Qinan Zhao, Xiaoshuan Zhang, Hailing Luo,

Improved composite deep learning and multi-scale signal features fusion enable intelligent and precise behaviors recognition of fattening Hu sheep, Computers and Electronics in Agriculture https://doi.org/10.1016/j.compag.2024.109635

Comment 2:

The article should delve more deeply into data privacy and security issues, especially when using IoT technologies to collect and transmit sensitive data.

Comment 3:

The article utilizes a literature review approach and should discuss the limitations of its research methodology, suggesting that the authors provide more detail on the method of literature selection.

Comment 4:

The article contains only one table and it is recommended that the author add charts and illustrations to help explain complex data and concepts. This is because clear and accurate schematic representations help in understanding the content of the article.

Reviewer 3 Report

Comments and Suggestions for Authors

The paper deals with a very interesting current issue. The authors provide interesting information concerning wearable collar technologies used in cattle farms, highlighting their transformative role in monitoring animal behaviour, health, and productivity. The references that appear are appropriate and up-to-date. The article seems coherent, the authors have processed it systematically, and it has a good concept, but I still have a few comments.

1- The authors have exhaustively reviewed the available collar technologies and their applications. Nevertheless, there are several limitations to these sensors' spread. Data validation is definitely an important aspect. A further issue is the economic efficiency and attendant return on investment. The manuscript could be improved on these aspects. Discussing these issues would offer a more comprehensive perspective and identify future challenges.

2- Section 5.1.2: The studies mentioned seem to be reported disconnectedly. I suggest reorganizing the section.

3- Section 5.2.2 and 5.3.2: For sensors that are not currently available, it could be interesting to report the limitations preventing or slowing down their entry into the market.

4-  Table 1: by what criterion is the table organized? In the absence of an author-defined criterion, it is recommended that the table be sorted according to the reported type of sensors to enhance its readability.

5-  The manuscript contains a unique table summarizing the sensors currently available on the market. To provide further support for the information contained within sections entitled 'Existing Commercial Solutions' (sections 5.1.2./ 5.4.2 /5.5.2 / 5.6.2), it would be useful to add new tables which present a comparison with relevant practical information, such as performance, energy consumption/efficiency, cost, reliability and distribution of the different solutions. Incorporating such tables would emphasize which solutions are most appropriate and chosen for on-farm applications.

6-  I suggest partially modifying the keywords. To optimize visibility, it is always recommended not to repeat words from the article's title.

7-        Line 284: There is a typo.

8-     Line 806, line 836: Please carefully check your references; some are not in the manuscript. 

Reviewer 4 Report

Comments and Suggestions for Authors

This manuscript provides a comprehensive review of wearable collar technologies for dairy cows, detailing their applications, current uses, and potential innovations within precision livestock farming. I appreciate the substantial effort the authors have put into this work. I understand the challenges of writing a literature review, which is by no means an easy task. However, the manuscript raises several concerns that need to be addressed by the authors:

  1. I recommend further summarizing the main findings of the review in the conclusion section, while explicitly highlighting directions for future research. For example, you could specify which technologies (e.g., accelerometers, GPS) show the most promise and identify key technical challenges that need to be addressed in future studies.

  2. Although the manuscript discusses the potential of various technologies, it lacks sufficient exploration of how different sensor technologies could be integrated into multifunctional systems. As we know from practical applications, a single collar often incorporates multiple sensors. This integration should be discussed in greater detail.

  3. When searching the literature, what was the starting and ending timeframe? For example, if it was from 2005 to 2022 (line 72), this appears to belong to the Materials and Methods section rather than being a general statement. Please clarify.

  4. Some authors limit their search to specific sub-databases, such as SCIE and SSCI within the Web of Science, to ensure peer-reviewed sources. Did you follow a similar approach, or was your search broader? Please provide details.

  5. The choice of keywords for the literature search seems somewhat unclear. Could you provide a more direct explanation or even display your search formula? Did you employ synonyms for key terms (e.g., "dairy cow" and "dairy cattle")? Exploring such synonyms could yield a broader set of relevant literature.

  6. While it is acceptable to heavily reference existing literature in a review, the manuscript appears to rely extensively on citations without providing sufficient original perspectives or in-depth analysis from the authors. This is a critical limitation.

I look forward to seeing a revised version of the manuscript that addresses these points.

Reviewer 5 Report

Comments and Suggestions for Authors

This paper presents a survey of collars used to monitor dairy cows. The paper is well-organized and easy to read. However, it lacks a comparison with other reviews to state its novelty clearly. In this sense, some of the topics are already covered in other reviews, for instance:

- Livestock feeding behaviour: A review on automated systems for ruminant monitoring

- Review: Precision livestock farming technologies in pasture-based livestock systems

- Farm 4.0: Innovative smart dairy technologies and their applications as tools for welfare assessment in dairy cattle

- A systematic literature review on the use of deep learning in precision livestock detection and localization using unmanned aerial vehicles

Moreover, the cited literature review should be added and updated. For example:

L126-137: I recommend using more recent works to highlight the difference in this work.

In section 5.4: there are several studies not mentioned in this section, such as:

- On-Device Feeding Behavior Analysis of Grazing Cattle

- A full end-to-end deep approach for detecting and classifying jaw movements from acoustic signals in grazing cattle

- Using segment-based features of jaw movements to recognise foraging activities in grazing cattle

- The implications of compound chew–bite jaw movements for bite rate in grazing cattle

- Discriminative power of acoustic features for jaw movement classification in cattle and sheep

Please, also consider this comment for sections 5.1 to 5.6

Other minor comments are:

L43: This comparison could be biased by the pandemic in 2020. Could it be possible to compare the milk production between another consecutive 2 years?

L215: It is not entirely clear that "sensor" refers not to accelerometers but to the accelerometer and the associated electronics.

I considered more properly talking about "sensor usability" than "sensor performance" in this context.

Round 2

Reviewer 1 Report

Comments and Suggestions for Authors

Overall, the authors have addressed the comments, and I am content with most of the responses. There are some outstanding below and some areas where more details are needed.

Lines 53-55: References are required where there are statements here e.g., environmental sustainability etc., and throughout the manuscript.

Lines 78-80: References needed here to support each of these. Unsure about the automatic changing of feed composition?

R13: Section 3 – I don’t think this edit provides a good reason for why collars were chosen for the review. There needs to be a clearer rationale here. There are ear-tag mounted systems available now and I would argue that there needs to be justification for not including these. You say that it adopts a more focused approach than previous reviews but if anything, it is broader. A couple of other weak areas also include ‘Collars were chosen because they offer a comfortable fit, ensuring minimal stress and disruption to the animal's natural behaviour. Furthermore, their positioning on the neck provides a stable platform for accurate sensor readings across various activities.’ I would say that ear tags provide this too.

Line 144: refers exclusively to collar-mounted…

Section 4

I don’t see any description of the type of review that was undertaken. If it is not a systematic review, what type is it?

Line 184: It would be useful to have reference to these 97 papers in a supplementary spreadsheet for example.

Line 185 – should the timespan be 1997 to 2024?

Section 5

Line 276: We still don’t know the breakdown of the performance here. Was it only grazing and non-grazing identified? Better with individual performance metrics for these.

Line 299: The reference to [57] should include a summary of what they found given that collars and leg accelerometers were used here.

Section 5.1.2

R28: Okay I understand your rebuttal on this.

What I would say is that anywhere is this section which states anything about performance requires a metric associated with it. For example, on line 326 where you say that “confirming that the device provides detailed information for herd management and disease prevention by accurately measuring rumination and oestrus activity…” we need to know the accuracy. There are many such example sin this section.

R29: Please see previous comment above.

R30: I think there is duplication in Section 5.2.2.

R35: Are there any papers that have assessed performance though? If so they should be discussed.

R37: You refer to heifers on line 600 in revised manuscript – ‘also accurate for heifers’ – this is another example of where performance is needed for context.

Author Response

Overall, the authors have addressed the comments, and I am content with most of the responses. There are some outstanding below and some areas where more details are needed.

R) The authors thank the Reviewer for the constructive comments and suggestions, increasing the quality of the paper.

1)           Lines 53-55: References are required where there are statements here e.g., environmental sustainability etc., and throughout the manuscript.

R1)        Thanks for the comment. We have added the relevant references in the proper position.

2)         Lines 78-80: References needed here to support each of these. Unsure about the automatic changing of feed composition?

R2)        The authors thank the Reviewer for the insightful suggestions. You are correct that the phrasing regarding the automatic adjustment of feed composition could lead to confusion. To address this, we have revised the sentence for clarity and precision. Additionally, we have added the appropriate references to support the statements throughout the section. The updated text now reads as follows:

"Digital interfaces allow farmers to visualise the collected data and integrate it with other animal-related information, generating specific insights regarding health, behaviour, production, and the environment, which support the decision-making process, either performed by the farmer or fully automated, based on scientific evidence and predictive data analysis [8,11,12]. This process enables proactive and individualized management, reducing the need for manual interventions—such as physically inspecting each cow for signs of illness, manually identifying cows in heat for breeding [8], or adjusting feed rations based on observed behaviour [12]. Instead, automated systems can detect early signs of disease through changes in rumination patterns, trigger alerts for optimal insemination timing based on activity data, and provide recommendations for adjusting feed composition based on real-time consumption metrics—significantly improving the overall efficiency of farm operations [8,11,12]."

3)           R13: Section 3 – I don’t think this edit provides a good reason for why collars were chosen for the review. There needs to be a clearer rationale here. There are ear-tag mounted systems available now and I would argue that there needs to be justification for not including these. You say that it adopts a more focused approach than previous reviews but if anything, it is broader. A couple of other weak areas also include ‘Collars were chosen because they offer a comfortable fit, ensuring minimal stress and disruption to the animal's natural behaviour. Furthermore, their positioning on the neck provides a stable platform for accurate sensor readings across various activities.’ I would say that ear tags provide this too.

R3)        The authors thank the Reviewer for the valuable feedback. We are aware about the importance of exploring alternative technologies, and we plan to address other technologies in future studies. This review, however, was specifically designed to concentrate on collars without excluding the relevance of other systems. We hope this clarifies our rationale. We have chosen to focus on collars for this review due to their widespread application and established role in precision livestock farming. While we recognize that ear-tag-mounted systems and other technologies offer comparable benefits, our intent was to adopt a focused approach to provide a comprehensive analysis of one device type with a detailed analysis of the current available devices on the market.

4)           Line 144: refers exclusively to collar-mounted…

R4)        Thank you for your observation. We appreciate your attention to the detail. To ensure clarity and precision, we have revised the text to explicitly refer to collar-mounted systems. The updated text now reads as follows:

"Gyroscopes and magnetometers, also mounted on the neck, provide additional information on angular movements and head orientation [28-35]. Additionally, vocalizations can be recorded by a microphone attached to the collar [36]. Cow identification can be effectively carried out using RFID tags integrated into the collar [37]."

Section 4

5)           I don’t see any description of the type of review that was undertaken. If it is not a systematic review, what type is it?

R5)        Thank you for your comment. We have clarified the type of review assumed in the manuscript. Following the well-established literature in the field (see the reference below for details), we proposed a systematized review. This specification has been added in the title and in the abstract of the revised paper to avoid any misunderstanding.

Grant, M.J.; Booth, A. A Typology of Reviews: An Analysis of 14 Review Types and Associated Methodologies. Health Inf. Libr. J. 2009, 26, 91–108. [CrossRef] [PubMed] https://onlinelibrary.wiley.com/doi/10.1111/j.1471-1842.2009.00848.x

6)           Line 184: It would be useful to have reference to these 97 papers in a supplementary spreadsheet for example.

R6)        The authors thank the Reviewer for their suggestion. After the first round of revisions, we clarified that the articles representing the most impactful studies for the review total 55. These papers are already clearly reported in the table provided within the manuscript. Furthermore, all these articles are listed in the bibliography, ensuring full transparency and accessibility. We believe that creating a supplementary spreadsheet file would be redundant, as the most relevant articles are already detailed comprehensively in the manuscript.

7)           Line 185 – should the timespan be 1997 to 2024?

R7)        Thank you for pointing this out. We have corrected the timespan in "1997 to 2025", because during the first round of revisions, we incorporated new more recent references, including articles published in the current year.

Section 5

8)           Line 276: We still don’t know the breakdown of the performance here. Was it only grazing and non-grazing identified? Better with individual performance metrics for these.

R8)        Thank you for your comment. We have provided a detailed breakdown of the accuracy, specifying the various behaviours considered within the distinct categories.

9)           Line 299: The reference to [57] should include a summary of what they found given that collars and leg accelerometers were used here.

R9)        Thank you for your insightful comment. We have updated the manuscript to include a summary of the findings from reference [57], as suggested. The revised text now reads as follows: "For instance, a study combining neck-mounted accelerometers on collars, leg-mounted accelerometers, and ultra-wide band (UWB) localization sensors found that this approach improved detection performance within 24–8 hours before calving. The neck-mounted accelerometer on the collar achieved a Precision of 50–53% and Sensitivity of 47–48% (AUC 86–88%), while combining sensors further enhanced detection performance [57]."

Section 5.1.2

10)        R28: Okay I understand your rebuttal on this.

What I would say is that anywhere is this section which states anything about performance requires a metric associated with it. For example, on line 326 where you say that “confirming that the device provides detailed information for herd management and disease prevention by accurately measuring rumination and oestrus activity…” we need to know the accuracy. There are many such examples in this section.

R10)      We agree with the Reviewer that accuracy values are important; however, a fundamental issue lies in the lack of uniformity in the definition of criteria, thresholds, and reference gold standards for validating collar-based sensors. This inconsistency makes comparisons challenging and prone to misinterpretation. Additionally, in most of the cited papers, different behavioural classes have been considered and analysed, making a rigorous and meaningful comparison impossible.

Furthermore, without a detailed description of each experimental test conducted, the accuracy values we could include would likely be of limited usefulness and beyond the scope of this paper. For the sake of clarity and conciseness, we have reported accuracy values only in cases where they provide an undeniable added value. We believe this approach also enhances the overall readability of the paper.

11)        R29: Please see previous comment above.

R11)      This review was specifically designed to focus on collars while acknowledging the relevance of other systems. We hope this clarifies our rationale and objectives. While we recognize the value provided by other reviews, our intent was to adopt a focused approach, delivering a comprehensive analysis of a single device type. This includes a detailed evaluation of the currently available devices on the market, conducted within the framework of a systematized review.

12)        R30: I think there is duplication in Section 5.2.2.

R12)      We have solved these issues, which were caused by an error in using the track changes mode in the MS Word file. Thank you for your advice.

13)        R35: Are there any papers that have assessed performance though? If so they should be discussed.

R13)      Unfortunately, among the papers included in the review, and to the best of the authors' knowledge, no studies have investigated or compared the results obtained using vocalization with other animal-based measures. The literature on vocalization cited in this review has already been extensively discussed in the paper.

14)        R37: You refer to heifers on line 600 in revised manuscript – ‘also accurate for heifers’ – this is another example of where performance is needed for context.

R14)      Thank you for your comment. We have provided details about accuracy, as suggested.

Reviewer 3 Report

Comments and Suggestions for Authors

Dear authors, I would like to thank you for your response to each comment and for implementing the requested changes.

I would like to draw your attention to the paragraph 5.2.2, in which there are repeated sentences that should be removed. In particular:

- Line 455 the same sentence is repeated at LINE 465.

-Line 456 the same sentence is repeated at LINE 466.

- Line 459 the same sentence is repeated at LINE 469. 

I have no further comments.

Author Response

Dear authors, I would like to thank you for your response to each comment and for implementing the requested changes.

R) The authors thank the Reviewer for the constructive comments and suggestions, increasing the quality of the paper.

I would like to draw your attention to the paragraph 5.2.2, in which there are repeated sentences that should be removed. In particular:

- Line 455 the same sentence is repeated at LINE 465.

-Line 456 the same sentence is repeated at LINE 466.

- Line 459 the same sentence is repeated at LINE 469.

I have no further comments.

R) We have fixed these problems due to an error in the use of the track changes mode in the MS Word file. Thanks for your advice.

Reviewer 4 Report

Comments and Suggestions for Authors

The authors have addressed all my concerns, and I believe this manuscript can be published.

Author Response

The authors have addressed all my concerns, and I believe this manuscript can be published.

R) The authors thank the Reviewer for the constructive comments and suggestions, increasing the quality of the paper.

Reviewer 5 Report

Comments and Suggestions for Authors

All my points have been solved

Author Response

All my points have been solved

R) The authors thank the Reviewer for the constructive comments and suggestions, increasing the quality of the paper.